# Deformation Inhomogeneities of a Hypoeutectic Aluminum-Silicon Alloy Modified by Electron Beam Treatment

**DOI:** 10.3390/ma16062329

**Published:** 2023-03-14

**Authors:** Artem Ustinov, Anatoly Klopotov, Yuri Ivanov, Dmitry Zagulyaev, Anton Teresov, Elizaveta Petrikova, Denis Gurianov, Andrey Chumaevskii

**Affiliations:** 1Faculty of Civil Engineering, Tomsk State University of Architecture and Building, Solyanaya Sq. 2, Tomsk 634003, Russia; ustinov_a_m@tsuab.ru (A.U.); klopotovaa@tsuab.ru (A.K.); 2Institute of High Current Electronics SB RAS, Akademicheskij Ave. 2/3, Tomsk 634055, Russia; yufi55@mail.ru (Y.I.); tad514@yandex.ru (A.T.); elizmarkova@yahoo.com (E.P.); 3Department of Natural Sciences Named after Professor V.M. Finkel, Siberian State Industrial University, Kirova Str. 42, Novokuznetsk 654007, Russia; zagulyaev_dv@physics.sibsiu.ru; 4Institute of Strength Physics and Materials Science, Siberian Branch of Russian Academy of Sciences, Akademicheskij Ave. 2/4, Tomsk 634055, Russia; desa-93@mail.ru

**Keywords:** electron beam treatment, aluminum-silicon alloy, deformation inhomogeneities, surface hardening

## Abstract

This paper presents the results of uniaxial tensile tests on specimens of the hypoeutectic aluminum-silicon alloy A319. According to the results, the influence of surface treatment by pulsed electron beam on the mechanical properties of the material was determined. The peculiarities of deformation localization in the material caused by grinding of the surface layer material structure due to rapid crystallization during electron beam treatment were revealed. The surface treatment up to the depth of 100 µm leads to the formation of a fine dendritic columnar structure of silumin and to an increase in the plasticity of the samples. The influence of the surface treatment affects the increase in the deformation localization in the region of the stable concentrator before failure. The greatest increase in ductility and localization of deformation occurs during treatment with an energy density of 15 J/cm^2^. In the process of specimen deformation, unstable, metastable, and stable areas of plastic deformation localization are formed and replaced, and the formation of stable areas of localized plastic deformation, in which the specimen fails at the end of the test, can be detected at the initial stages of testing. In specimens, during the test in the zone of localized plastic deformation, bands are formed which pass through the entire surface of the specimen at an angle of 35 to 55 degrees to the tensile axis, and their development leads to the formation of stable zones of localized plastic deformation and to the failure of the specimen.

## 1. Introduction

Changing the surface properties of structural materials can lead to an increase in the strength or plasticity of the product as a whole. Elimination of surface layer defects, refinement of grain structure, formation of optimal structural phase composition from the point of view of plastic deformation and fracture localization peculiarities can contribute to change the plastic deformation and fracture characteristics. High-energy pulsed electron beam is one of the promising tools for modifying the surface layer of metals and alloys [1]. As a result of electron beam treatment, better technological properties are provided in comparison with traditional methods of surface treatment, such as mechanical, thermal, or thermo-mechanical treatment [2,3].

A number of research groups are studying the behavior of the structure, defect substructure, elemental and phase composition [4,5,6,7], improvement of physical and mechanical properties [8], and enhancement of corrosion resistance [9] of metallic materials treated with pulsed electron beam.

Several studies [10,11] have focused on the effect of high energy beam (200 keV) on the deformation behavior of Al-4Cu alloy. Studies have shown that electron-beam treatment can affect the deformability and strain rate sensitivity in a small volume of aluminum alloy by changing the viscous flow of a superfine amorphous membrane composed of natural oxide on the surface.

In works [1,12] it is shown that treatment of silumin by pulsed electron beam is accompanied by formation in a near-surface layer of submicro- and nanocrystalline columnar structure as a result of superfast heating and cooling. The formation of such micro- and nanoscale multiphase structure and the grinding of large silicon wafers to the nanoscale state are the main reasons for the multiple increase in fatigue life [13] and strength properties [2] of electron beam treated silumin. As a result, the near-surface layer obtained by high-energy treatment can be considered as an independent functional subsystem, which has a significant influence on the scale levels of material plastic flow localization [14].

A brief review of modern studies of the effect of electron beam treatment on the deformation behavior of metals allows us to conclude that this issue is a promising area of research and requires further development. At the same time, it is still unknown how a change in the structure and properties of a thin surface layer of the material can contribute to a change in the characteristic features of deformation localization and fracture of the samples as a whole. Therefore, the purpose of the present work is to study the influence of pulsed electron beam surface treatment of A319 grade hypoeutectic silumin on the characteristics of its plastic deformation and fracture under tensile stress.

## 2. Materials and Methods

The samples were obtained from a hypoeutectic aluminum-silicon alloy of A319 grade (Al-(4-6)Si-1.3Fe-0.5Mn-0.5Ni-0.2Ti-2.3Cu-0.8Mg-1.5Zn, wt%). Samples in the form of bilateral proportional blades with the working area dimensions of 55 mm × 9.6 mm × 3.8 mm were prepared (Figure 1a). The samples were cut on a DK7750 electric discharge machine. The surface modification of the samples was performed according to the scheme shown in Figure 1b. The samples were processed from both sides by pulsed electron beam at the SOLO (Institute of High Current Electronics) setup [15,16] with the following parameters: accelerated electron energy 18 keV, electron beam pulse duration 150 µs, pulse number 3, pulse repetition frequency 0.3 s^−1^, electron beam energy density 15 and 25 J/cm^2^. As shown in [3], these treatment modes lead to rapid melting of the silumin surface layer and differ in the thickness of the molten layer.

Tensile testing of the specimens was performed on an Instron 3386 testing machine at a tensile speed of 0.2 mm per minute. The test scheme is shown in Figure 1c. The grips of the testing machine provided reliable specimen clamping and centering during the tests. The distribution of strain fields on the surface of the specimens during uniaxial tension was obtained using the Vic-3D digital optical system based on the method of stereo digital image correlation [17]. Before the tests, speckle images were painted on the specimen surfaces [18].

Measurements of the development of deformation fields on the surface of the investigated specimens were carried out using the VicSnap program on the basis of the obtained synchronous recording of images from two cameras (Figure 1c). As a result, the fields of relative longitudinal deformation (ε_yy_—along the Y axis, axis directed along the applied stress), transverse deformation (ε_xx_—along the X axis, axis directed across the applied stress) were determined and analyzed [19,20]. The deformation fields were analyzed at different stages of the test process up to the pre-fracture stage. After the tests, the fracture structure was examined using a Zeiss LEO EVO 50 scanning electron microscope in the secondary electron mode.

## 3. Results

Electron beam treatment of the surface causes rapid melting of the surface layer and subsequent crystallization at a sufficiently high rate. This results in the refinement of the alloy structure in the thin surface layer and the formation of a peculiar morphology on the surface. This, in turn, determines the characteristics of deformation and fracture of the material during testing. Figure 2 shows photographs of specimens after testing. All three types of specimens are characterized by fracture with a low degree of plastic deformation without “necking”.

The plastic deformation of the specimens under test is fairly uniform until ultimate strength and fracture are reached. Figure 3 shows the uniaxial tensile test plots in coordinates σ (normal stresses along the tensile axis) of <ε_yy_> (total strain in the work area) of untreated specimens and specimens treated with 15 and 25 J/cm^2^ electron beam energy density (ES).

Electron beam treatment of the surface does not result in significant changes in tensile strength. The maximum stress before failure of the specimens varies within 4%. However, analysis of the change in strain of the specimens during the test shows that the plasticity of the specimens changes quite significantly. Maximum relative strains of specimens treated by pulsed electron beam with energy density 15 J/cm^2^ increased by 55%, and with ES = 25 J/cm^2^ by 27%. The maximum relative strains of the untreated specimen were 0.234%, at ES = 25 J/cm^2^ were 0.297%, and at ES = 15 J/cm^2^ were 0.363%. The Young’s modulus of the untreated and electron beam-treated samples at ES = 25 J/cm^2^ is the same and is equal to 60 GPa, and at ES = 15 J/cm^2^ it decreases by 3.3% and is equal to 58 GPa. The tangent modulus of deformation at the point of failure of the specimen without treatment was 15 GPa. For the specimens treated with ES = 15 J/cm^2^ and 25 J/cm^2^, it decreased by 47 and 27%, respectively, which is equal to 8 and 11 GPa.

The increase in plasticity of samples treated by pulsed electron beam is caused by a significant change in the structure of the surface layer of silumin. The conducted studies show that in the cast state the structure of silumin A319 is represented by dendrites of solid solution based on aluminum, Al-Si eutectics, inclusions of silicon and needle-shaped intermetallides, which is clearly marked on the fracture surface (Figure 4a). The deformation of the specimen is initially accompanied by brittle fracture along the boundaries of the silicon inclusions, which initiates the fracture of the specimen as a whole (Figure 4b).

Treatment of silumin with a pulsed electron beam results in melting of the surface layer up to 15–100 µm thick (Figure 5a,c). High-speed crystallization and subsequent cooling due to heat dissipation into the inner volume of the material is accompanied by the formation of a finely dendritic cellular (columnar) structure in the surface layer, whose crystallite sizes vary within the range of (550–700) nm, which determines a different character of deformation and material fracture (Figure 5b,d). It can be clearly seen that the inclusions of second phases of needle (lamellar) shape, characteristic for the structure in the raw state, are not detected in this layer by scanning electron microscopy methods. This is one of the main reasons for the increase in plasticity of silumin after treatment. Treatment with energy 25 J/cm^2^ (Figure 5c,d) leads to formation of a surface layer with greater thickness and smaller size of dendrite arms in comparison with treatment with energy 15 J/cm^2^ (Figure 5a,b).

To analyze the stress-strain state of material specimens under uniaxial tension, the images of relative longitudinal <ε_yy_> and transverse <ε_xx_> strain distributions on the specimen surface were selected. The images show the working part of the specimen with the relative strain field plotted on it in shades of gray, with the scale indicated on the right. An extensometer (virtual strain gauge) was used to determine the overall deformed state of the specimen along the edges of the work area. Since the stress-strain diagram (Figure 3) of the untreated silumin specimen has no characteristic fractures, straight lines, squares, etc., the division into four stages is made conditionally. The curve is divided into four sections, at the end points of which the deformation patterns of the sample surface are shown (Figure 6).

The plastic deformation of the untreated sample involves the formation of a rather complex pattern of deformation fields with a chaotic distribution of deformation localization areas in the initial stage and the formation of stable deformation concentrators later (Figure 6). At a strain rate of 0.034%, areas of high and low relative strain are chaotically distributed, alternating in a staggered pattern (Figure 6a). As the strain rate increases to 0.077%, the deformation pattern becomes somewhat more complex (Figure 6b). The deformation develops in a part of concentrators with the formation of some similarity of bands between them, which are partially traceable and at the initial stage of deformation (1, 2 on Figure 6). According to their location on the specimen surface (35–55 degrees relative to the deformation axis) they can be attributed to Chernov-Luders bands. Another part of the concentrators does not show any development or disappears in the deformation field image.

The evolution of the deformation process up to 0.134% leads to an increase in deformation in the main deformation concentrators and a weakening of most of the others (Figure 6c). At the 0.233% pre-fracture stage, the presence of the main deformation concentrator is clearly evident (Figure 6d). This area of deformation localization is clearly identifiable from the early stages of the test (Figure 6). Thus, the test distinguishes: basic or stable deformation concentrators present in the deformation field patterns throughout the test; metastable deformation localization zones present in two or three stages of deformation; and unstable deformation concentrators appearing and disappearing only in one of the stages.

Plastic deformation of samples treated with 15 J/cm^2^ energy in the early stages of deformation shows a fairly similar pattern of distribution of localized plastic deformation zones (Figure 7a,b). A large number of unstable deformation concentrators are also formed within the same stage (E, F in Figure 7). Metastable concentrators are formed within two or three stages of deformation and disappear later (C, D in Figure 7). The largest zones of localized plastic deformation form no more than two or three, the action of the largest of which leads to the failure of the specimen (A,B in Figure 7a–d). Similarly to the previously described case, localized deformation bands are formed at an angle of 45 degrees to the deformation axis (1, 2 in Figure 7). Two types of bands are formed, and the largest deformation concentrators are formed at their intersection.

As the treatment energy of the specimens increases, the localized plastic deformation bands become more pronounced, starting from small stages of deformation (1, 2 in Figure 8). At the first stage, the zones of localized deformation are more chaotic and the bands are quite weak, but at the second and third stages their number increases sharply (Figure 8a–c). The development of a stable deformation concentrator in the central part of the specimen at the intersection of the two deformation bands leads to the failure of the specimen at the end of the test (Figure 8d).

It should be noted that the strain localization in the pre-fracture region is quantitatively inferior compared to the lower energy treated specimen. This may also be responsible for the lower overall ductility for this type of specimen. This situation can be attributed to the increased thickness of the treated layer.

The data obtained show significant differences in the deformation behavior of base metal specimens and specimens treated with electron beams of different energies. At the stage of pre-fracture, the samples of the raw material and the samples treated with different energies of the electron beam can be clearly distinguished both by the values of the longitudinal deformations (Figure 9) and by the transverse deformations (Figure 10). The values of maximum longitudinal deformations <ε_yy_> on the surface of the silumin sample without irradiation were 0.542%; at energy density ES = 15 J/cm^2^ the deformation values increased by 83% and were equal to 0.995%; at energy density ES = 25 J/cm^2^ the deformation values increased by 17% and were equal to 0.635%.

The increase in plasticity (+55%) of silumin samples at electron beam treatment with energy density 15 J/cm^2^ due to values of maximum longitudinal deformations <ε_yy_> (+83%) over the sample surface (relative to the raw sample). The increase in plasticity (+27%) of silumin samples at electron beam treatment with energy density 25 J/cm^2^ due to both increased values of maximum longitudinal deformations <ε_yy_> (+17%) and increased values of minimum deformations (+10%) across the sample surface (relative to the raw sample without treatment).

Values of maximum transverse deformations <ε_xx_> on a surface of silumin sample without treatment made 0.282%; at energy density ES = 15 J/cm^2^ values of deformations have increased on 89% and are equal to 0.535%; at energy density ES = 25 J/cm^2^ values of deformations have increased on 5% and are equal to 0.296%. The increase in ductility (+55%) of silumin specimens at electron beam treatment with energy density ES = 15 J/cm^2^ is due both to high values of maximum transverse strains <ε_xx_> (+89%) along the specimen surface (in comparison with the starting specimen) and to increased values of minimum strains (+82%) along the specimen surface (relative to the values of the starting specimen without treatment). The smaller increase in ductility (+27%) of silumin specimens at electron beam treatment with energy density ES = 25 J/cm^2^ is due to both smaller increase in values of maximum transverse deformations <ε_xx_> (+5%) and smaller increase in values of minimum deformations (+23%) over the specimen surface (in comparison with the raw specimen).

## 4. Conclusions

The conducted studies show that by treating the surface of hypoeutectic aluminum alloy A319 with pulsed electron beam it is possible to form in the surface layer a structure of the fine dendritic cell (columnar) type without inclusion of secondary phases of needle (lamellar) shape, which leads to a significant increase in plasticity of the material with a small change in strength properties. The depth of the treated zone varies from 15 to 100 μm. The greatest depth of the surface layer is characteristic of samples treated at 25 J/cm^2^. However, the greatest ductility is characteristic of the specimens treated with an electron beam with an energy of 15 J/cm^2^.

The obtained data testify to the localization of deformation in all types of specimens during testing. This process begins with the formation of a chaotic or staggered distribution of localized areas of plastic deformation. Further development of the plastic flow leads to the formation of deformation bands from individual concentrators located at an angle of 45 degrees to the deformation axis. Bands of two orientations intersecting each other are present in the specimens. Areas of maximum localization of plastic deformation are formed at the intersection of such bands. The tendency to band formation increases after surface treatment with electron beam, and the deformation localization is maximum in specimens treated with electron beam of energy 15 J/cm^2^, while the deformation bands are most pronounced in specimens treated with energy 25 J/cm^2^. Plastic deformation localization zones can be of stable, metastable, and unstable types. Zones of the first type form from small amounts of deformation and develop throughout the test. Zones of the second type may develop over a long period of time, but their effects then diminish or cease. Third type zones form for a short period of time and then disappear. The fracture structure of the base metal of treated and untreated specimens is represented by brittle chipping along the boundaries of the silicon inclusions. The destruction of the treated surface layer is characterized by intergranular fracture along the boundaries of columnar dendrite arms.

Thus, the results obtained demonstrate the effectiveness of electron beam surface treatment of aluminum-silicon alloys in reducing their brittleness and increasing their ductility.

## Figures and Tables

**Figure 1 materials-16-02329-f001:**
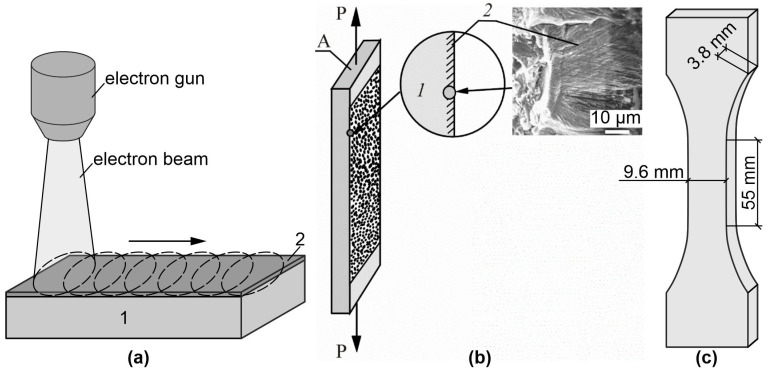
Scheme of specimen treatment (**a**), registration of speckle images from the modified surface of the specimen by pulsed electron beam treatment during tensile deformation (**b**), and dimensions of the tested specimens (**c**). The insets show schematic cross sections of the near surface regions of the samples. Inset (**b**) shows a microphotograph of the treated near-surface layer of an alloy: A—silumin plate; P—applied stress. 1—Base metal; 2—Electron beam treated near-surface layer.

**Figure 2 materials-16-02329-f002:**
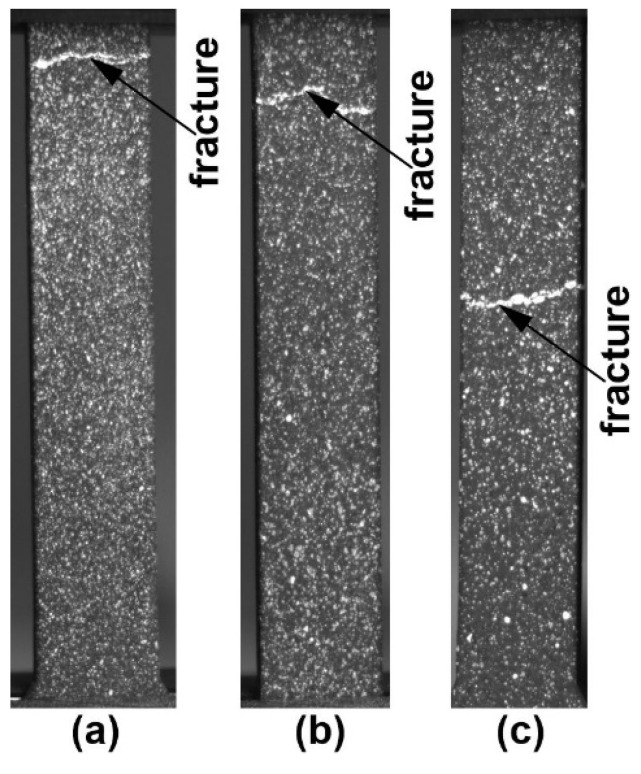
Photographs of working parts of flat silumin samples after tests ((**a**)—untreated) and treated with different densities of electron beam energy ((**b**)—15 J/cm^2^; (**c**)—25 J/cm^2^).

**Figure 3 materials-16-02329-f003:**
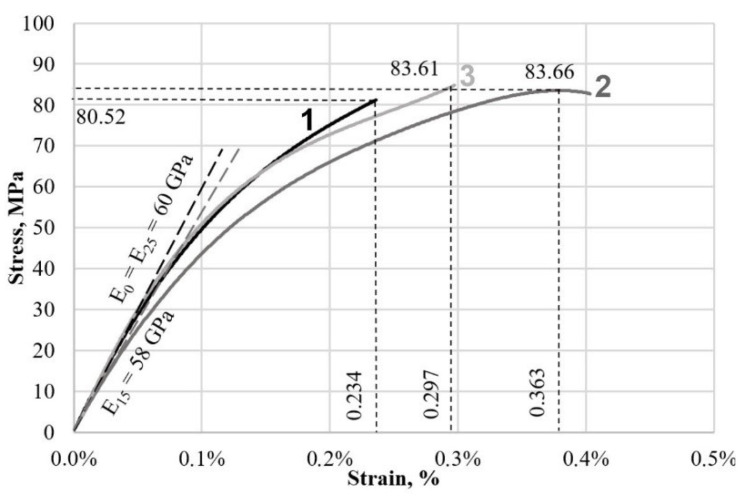
Smoothed strain diagrams of uniaxial tension of silumin with maximum values of stresses and strains 1—base metal; 2—sample treated at 15 J/cm^2^, 3—sample treated at 25 J/cm^2^.

**Figure 4 materials-16-02329-f004:**
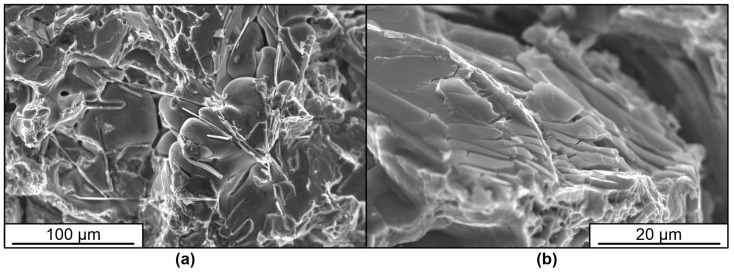
Fracture surface of silumin A319 in the raw state. Inclusions of silicon and acicular intermetallic compounds on the fracture surface (**a**), boundaries of silicon inclusions (**b**).

**Figure 5 materials-16-02329-f005:**
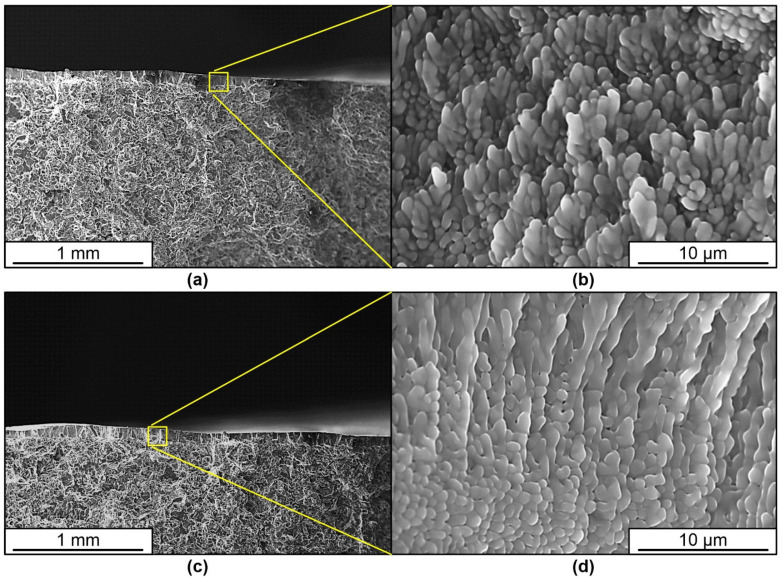
Fractography of the fracture surface of silumin A319 treated with a pulsed electron beam at an electron beam energy density of 15 J/cm^2^ (**a**,**b**) and 25 J/cm^2^ (**c**,**d**). The arrow indicates the treated area.

**Figure 6 materials-16-02329-f006:**
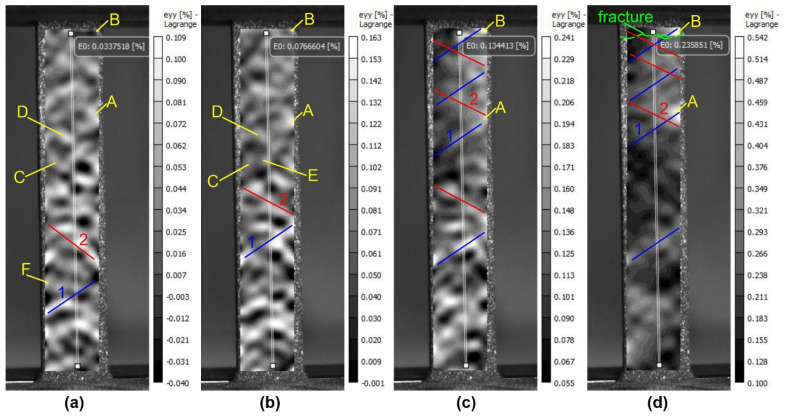
Patterns of the strain fields on the surface of an untreated silumin sample at total strain values; (**a**) 0.034%; (**b**) 0.077%; (**c**) 0.134%; (**d**) 0.233%: 1,2—bands of localized plastic deformation, A,B—stable, C,D—metastable, E,F—unstable areas of localized plastic deformation.

**Figure 7 materials-16-02329-f007:**
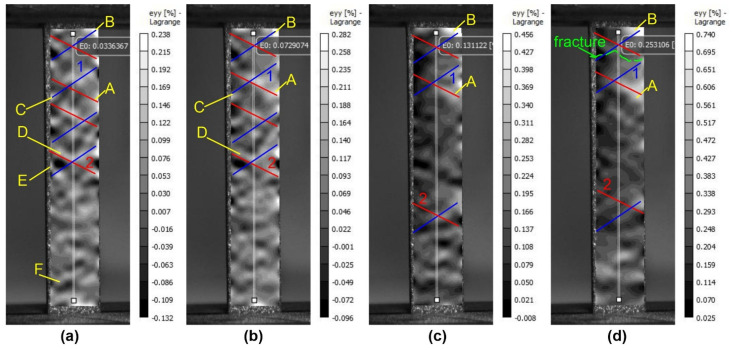
Patterns of strain fields on the surface of a treated silumin sample ES = 15 J/cm^2^ at the value of total strains (**a**) 0.0336%; (**b**) 0.0729%; (**c**) 0.1312%; (**d**) 0.2531%: 1,2—bands of local plastic deformation, A,B—stable, C,D—metastable, E,F—unstable areas of local plastic deformation.

**Figure 8 materials-16-02329-f008:**
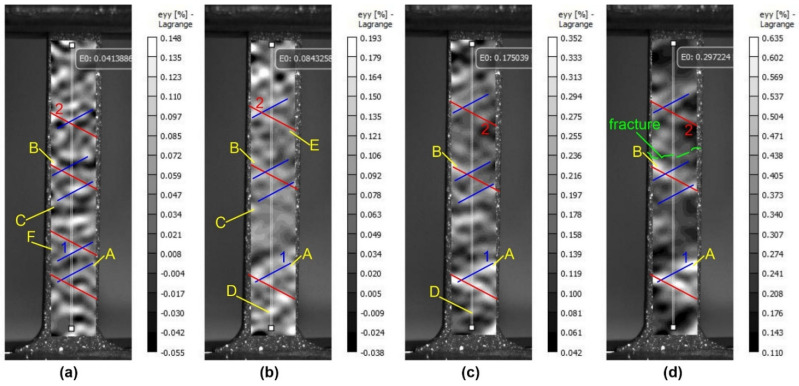
Patterns of strain fields on the surface of a treated silumin sample ES = 25 J/cm^2^ at the value of total strain (**a**) 0.041%; (**b**) 0.084%; (**c**) 0.175%; (**d**) 0.297%: 1,2—bands of local plastic deformation, A,B—stable, C,D—metastable, E,F—unstable areas of local plastic deformation.

**Figure 9 materials-16-02329-f009:**
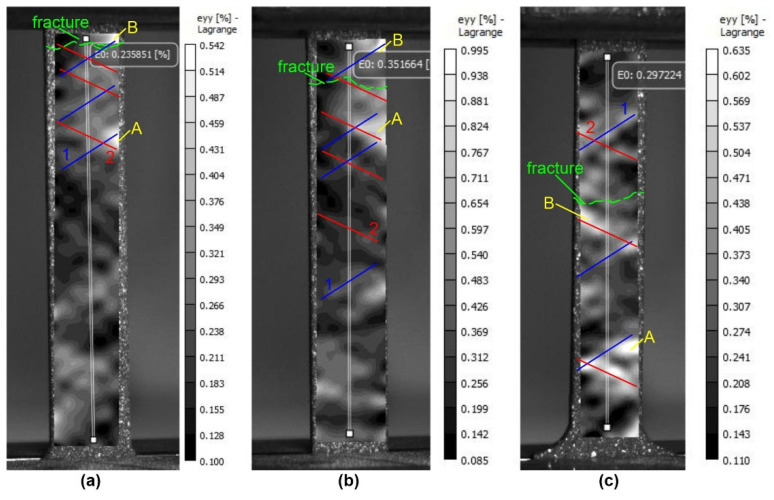
Patterns of distributions of relative longitudinal deformations <ε_yy_> on the surface of the silumin samples under uniaxial tension near the point of fracture: (**a**)—untreated silumin sample; (**b**)—silumin plate treated with an electron beam with energy density of 15 J/cm^2^; (**c**)—silumin plate treated with an electron beam with energy density of 25 J/cm^2^: 1,2—bands of localized plastic deformation, A,B—stable areas of localized plastic deformation.

**Figure 10 materials-16-02329-f010:**
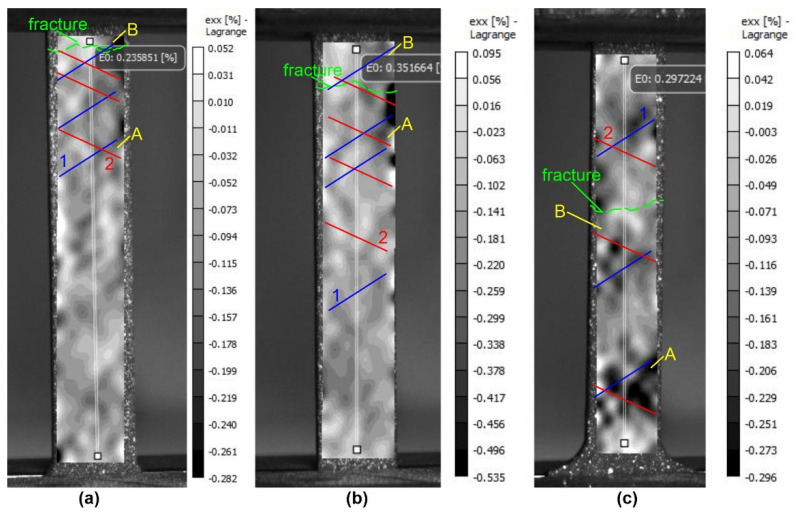
Distribution patterns of relative transverse strain <ε_xx_> on the surface of the silumin samples under uniaxial tension near the fracture point: (**a**)—untreated silumin sample; (**b**)—silumin plate treated with an electron beam with energy density of 15 J/cm^2^; (**c**)—silumin plate treated with an electron beam with energy density of 25 J/cm^2^: 1,2—bands of local plastic deformation, A,B—stable areas of local plastic deformation.

## Data Availability

Not applicable.

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
