# Peer review of "Deformation Inhomogeneities of a Hypoeutectic Aluminum-Silicon Alloy Modified by Electron Beam Treatment"

_materials, 2023, doi:10.3390/ma16062329_

Round 1

Reviewer 1 Report

This paper is on Deformation inhomogeneities of a hypoeutectic aluminum- 2 silicon alloy modified by electron beam treatment.  This is a good area of research and it need to be improved always. As per the need, authors done the work nicely and presented the results. It can be accepted for publication with some typological correction.

Author Response

Deat Mr./Ms reviewer

Thank you very much for your comments on the article.

Below are the responses to your comments.

Best regards

This paper is on Deformation inhomogeneities of a hypoeutectic aluminum- 2 silicon alloy modified by electron beam treatment.  This is a good area of research and it need to be improved always. As per the need, authors done the work nicely and presented the results. It can be accepted for publication with some typological correction.

A: Thank you for your comments. The article contains a number of corrections and additions.

Authors team

Reviewer 2 Report

In my opinion experimental results are interesting and seem useful for readers of Materials journal. The paper contains 10 figures – figures are legible and in most good quality.

Defect of paper is mostly unclear presentation and analysis diminish considerably the message and the usefulness of the paper. The main weakness of the paper is the lack of coherence throughout the manuscript. There are many confusing sentences with a poor clarity, these making the paper difficult to be read. I am asking for corrections by a native speaker.

Introduction chapter should be correct. Authors should include new information about topic of a paper. Papers cited in the References 7 from all 20 are older then 5 years (35 %) of all cited papers. I propose review literature to add new papers (from the last 5 years) to references. Minimum 8 papers (40 %) are wrote by Authors. I have the impression that the paper is based on old information and research conducted in the Authors' scientific units. Author should include several modern papers of global research in this field – more information based on worldwide (global) study – not mostly from Russia.

I recommend the authors to completely rewrite the paper before resubmitting it again for publication. The manuscript should be rejected in current form.

Author Response

Deat Mr./Ms reviewer

Thank you very much for your comments on the article.

Below are the responses to your comments.

Best regards

Authors team

In my opinion experimental results are interesting and seem useful for readers of Materials journal. The paper contains 10 figures – figures are legible and in most good quality.

Defect of paper is mostly unclear presentation and analysis diminish considerably the message and the usefulness of the paper. The main weakness of the paper is the lack of coherence throughout the manuscript. There are many confusing sentences with a poor clarity, these making the paper difficult to be read. I am asking for corrections by a native speaker.

Introduction chapter should be correct. Authors should include new information about topic of a paper. Papers cited in the References 7 from all 20 are older then 5 years (35 %) of all cited papers. I propose review literature to add new papers (from the last 5 years) to references. Minimum 8 papers (40 %) are wrote by Authors. I have the impression that the paper is based on old information and research conducted in the Authors' scientific units. Author should include several modern papers of global research in this field – more information based on worldwide (global) study – not mostly from Russia.

I recommend the authors to completely rewrite the paper before resubmitting it again for publication. The manuscript should be rejected in current form.

A: Thank you very much for your comments. The article contains a number of corrections and additions.

The article has been edited for English language and content. Discussion of results and conclusions were added.

Added a discussion of the literature for the last 5 years:

The article [21] is a summary of topics related to the production and surface treatment of metals and alloys using electron-beam technologies. The advantages of these technologies as well as their combination with other methods are discussed. In papers [5, 22 - 24] attention is paid to the surface modification of AISI 1045, 304 steel by means of pulsed electron beam and the mechanism of microstructure modification and corrosion improvement of steel is investigated. In works [25 - 27] the possibility of removing metallic and nonmetallic impurities from the technogenic material by single and double refining under electron-beam melting conditions was studied. Widespread interest in the technology of processing by high-energy pulsed electron beam confirms its prospects.

  1. [Fu, Y., Hu, J., Zhang, X., Huo, W., Cao, X., Zhao, W. View / Surface modification of AISI 1045 steel by pseudospark based pulsed electron beam // Nuclear Instruments and Methods in Physics Research, Section B: Beam Interactions with Materials and Atoms. Volume 434, 1 November 2018, Pages 88-92 DOI: 10.1016/j.nimb.2018.08.023]
  2. [ Yulei Fu, Jing Hu, Wansheng Zhao, Fujun Peng, Weijie Huo, Xiaotong Cao Microstructure modification and corrosion improvement of AISI1045 steel induced by pseudospark electron beam treatment // Nuclear Instruments and Methods in Physics Research, Section B: Beam Interactions with Materials and Atoms. Volume 469, 15 April 2020, Pages 10-18. DOI: 10.1016/j.nimb.2020.02.033]
  3. [Yulei Fu, Jing Hu, Weijie Huo, Xiaotong Cao, Ruixue Zhang, Wansheng Zhao Characterization of High-current Pulsed Electron Beam Interaction with AISI 1045 Steel and the Microstructure Evolution // Procedia CIRP. Volume 68, 2018, Pages 196-199. doi:10.1016/j.procir.2017.12.047]
  4. [Study of the Possibility of Recycling of Technogenic Hafnium during Electron Beam Refining Katia Vutova, Vladislava Stefanova, Martin Markov, Vania Vassileva // Materials (Basel)/. 2022 Nov 29;15(23):8518. doi: 10.3390/ma15238518. ]
  5. [Behaviour of Impurities during Electron Beam Melting of Copper Technogenic Material Katia Vutova, Vladislava Stefanova, Vania Vassileva, Milen Kadiyski // Materials (Basel). 2022 Jan 26;15(3):936. doi: 10.3390/ma15030936.]
  6. [Recycling of Technogenic CoCrMo Alloy by Electron Beam Melting / Katia Vutova, Vladislava Stefanova ,Vania Vassileva and Stela Atanasova-Vladimirova // Materials 2022, 15(12), 4168; https://doi.org/10.3390/ma15124168]
  7. [Electron-Beam Surface Treatment of Metals and Alloys: Techniques and Trends / Stefan Valkov, Maria Ormanova and Peter Petrov // Metals 2020, 10(9), 1219; https://doi.org/10.3390/met10091219]

Reviewer 3 Report

The manuscript discusses the deformation mechanism and inhomogeneities of electron beam modified Al-Si hypoeutectic alloys.

The specimens were treated by a pulsed electron beam where the energy density has been chosen 15 and 25 J/cm2.

The treatment procedure leads to the formation of a fine dendritic structure and improvement in plasticity.

The specimen treated with an energy density of 15 J /cm2 has higher ductility than that of 25 J/cm2.

Overall, the paper is very interesting and is worthy of publication.

The authors can improve the discussion of the results obtained.

For example, why the strain localization in the pre-fracture region of the specimen treated by the higher energy density is quantitatively inferior and less ductile?

Author Response

Deat Mr./Ms reviewer

Thank you very much for your comments on the article.

Below are the responses to your comments.

Best regards

Authors team

The manuscript discusses the deformation mechanism and inhomogeneities of electron beam modified Al-Si hypoeutectic alloys.

The specimens were treated by a pulsed electron beam where the energy density has been chosen 15 and 25 J/cm2.

The treatment procedure leads to the formation of a fine dendritic structure and improvement in plasticity.

The specimen treated with an energy density of 15 J /cm2 has higher ductility than that of 25 J/cm2.

Overall, the paper is very interesting and is worthy of publication.

The authors can improve the discussion of the results obtained.

For example, why the strain localization in the pre-fracture region of the specimen treated by the higher energy density is quantitatively inferior and less ductile?

A: Thank you very much for your comments. The article contains a number of corrections and additions.

The main reason for the localization of strain in the area of pre-fracture on the surface of the sample treated with higher energy density and lower ductility is the appearance of microcracks at the grain boundaries (Fig. 5 (b)).

Reviewer 4 Report

The paper entitled „Deformation inhomogeneities of a hypoeutectic aluminum-silicon alloy modified by electron beam treatment” focuses on studying the influence of pulsed electron beam surface treatment of A319 grade hypoeutectic silumin on the characteristics of its plastic deformation and fracture under tensile stress. The paper is well-written and interesting. Although the results are understandably submitted, the introduction and experimental parts should be improved.

I would like to recommend the publication of the paper publication after some changes concerning the following issues: 

1.               The introduction should be improved. More information about the importance and application of electron beam treatment of Al-Si alloys should be given.

2.               What is the initial structure of the aluminum-silicon alloy? Is it wrought or annealed? What was the average grain size? These characteristics refer to the depth of the EBT surface and the following structural changes. Therefore, such information together with a suitable micrograph should be added;

3.               The structure of the re-melted surface of the samples should be also shown. The difference in the grain size in the surface area could be also commented on.

4.               Measurements of the in-depth distribution of microhardness should be also presented.

5.               The conclusion should re-formulated. It sounds more like a summary of the results rather than a conclusion.

6.               The reference style does not meet the requirements of the journal. DOI is also missing.

Author Response

Deat Mr./Ms reviewer

Thank you very much for your comments on the article.

Below are the responses to your comments.

Best regards

Authors team

The paper entitled „Deformation inhomogeneities of a hypoeutectic aluminum-silicon alloy modified by electron beam treatment” focuses on studying the influence of pulsed electron beam surface treatment of A319 grade hypoeutectic silumin on the characteristics of its plastic deformation and fracture under tensile stress. The paper is well-written and interesting. Although the results are understandably submitted, the introduction and experimental parts should be improved.

I would like to recommend the publication of the paper publication after some changes concerning the following issues: 

  1. The introduction should be improved. More information about the importance and application of electron beam treatment of Al-Si alloys should be given.

A: Thank you very much for your comments. The introduction has been expanded to include a discussion of the current literature on electron beam processing.

The article [21] is a summary of topics related to the production and surface treatment of metals and alloys using electron-beam technologies. The advantages of these technologies as well as their combination with other methods are discussed. In papers [5, 22 - 24] attention is paid to the surface modification of AISI 1045, 304 steel by means of pulsed electron beam and the mechanism of microstructure modification and corrosion improvement of steel is investigated. In works [25 - 27] the possibility of removing metallic and nonmetallic impurities from the technogenic material by single and double refining under electron-beam melting conditions was studied. Widespread interest in the technology of processing by high-energy pulsed electron beam confirms its prospects.

  1. [Fu, Y., Hu, J., Zhang, X., Huo, W., Cao, X., Zhao, W. View / Surface modification of AISI 1045 steel by pseudospark based pulsed electron beam // Nuclear Instruments and Methods in Physics Research, Section B: Beam Interactions with Materials and Atoms. Volume 434, 1 November 2018, Pages 88-92 DOI: 10.1016/j.nimb.2018.08.023]
  2. [ Yulei Fu, Jing Hu, Wansheng Zhao, Fujun Peng, Weijie Huo, Xiaotong Cao Microstructure modification and corrosion improvement of AISI1045 steel induced by pseudospark electron beam treatment // Nuclear Instruments and Methods in Physics Research, Section B: Beam Interactions with Materials and Atoms. Volume 469, 15 April 2020, Pages 10-18. DOI: 10.1016/j.nimb.2020.02.033]
  3. [Yulei Fu, Jing Hu, Weijie Huo, Xiaotong Cao, Ruixue Zhang, Wansheng Zhao Characterization of High-current Pulsed Electron Beam Interaction with AISI 1045 Steel and the Microstructure Evolution // Procedia CIRP. Volume 68, 2018, Pages 196-199. doi:10.1016/j.procir.2017.12.047]
  4. [Study of the Possibility of Recycling of Technogenic Hafnium during Electron Beam Refining Katia Vutova, Vladislava Stefanova, Martin Markov, Vania Vassileva // Materials (Basel)/. 2022 Nov 29;15(23):8518. doi: 10.3390/ma15238518. ]
  5. [Behaviour of Impurities during Electron Beam Melting of Copper Technogenic Material Katia Vutova, Vladislava Stefanova, Vania Vassileva, Milen Kadiyski // Materials (Basel). 2022 Jan 26;15(3):936. doi: 10.3390/ma15030936.]
  6. [Recycling of Technogenic CoCrMo Alloy by Electron Beam Melting / Katia Vutova, Vladislava Stefanova ,Vania Vassileva and Stela Atanasova-Vladimirova // Materials 2022, 15(12), 4168; https://doi.org/10.3390/ma15124168]
  7. [Electron-Beam Surface Treatment of Metals and Alloys: Techniques and Trends / Stefan Valkov, Maria Ormanova and Peter Petrov // Metals 2020, 10(9), 1219; https://doi.org/10.3390/met10091219]

  1. What is the initial structure of the aluminum-silicon alloy? Is it wrought or annealed? What was the average grain size? These characteristics refer to the depth of the EBT surface and the following structural changes. Therefore, such information together with a suitable micrograph should be added;

A: Structural studies have been added to the article

The structure of the raw cast silumin A319. Samples were cut from the cast billet on a DK7750 electric discharge machine. The samples were obtained from a hypoeutectic aluminum-silicon alloy of A319 grade (Al-(4-6)Si-1.3Fe-0.5Mn-0.5Ni-0.2Ti-2.3Cu-0.8Mg-1.5Zn, wt%) with grain size from 45 to 65 μm.

  1. The structure of the re-melted surface of the samples should be also shown. The difference in the grain size in the surface area could be also commented on.

A: Structural studies have been added to the article

  1. Measurements of the in-depth distribution of microhardness should be also presented.

A: Data has been added to the article

Figure 6 shows the microhardness profiles of A319 silumin processed with an electron beam pulse duration of 200 µs.

  1. The conclusion should re-formulated. It sounds more like a summary of the results rather than a conclusion.

A: The conclusion has been revised

  1. Conclusion

The researches carried out show that by processing the surface of the hypoeutectic aluminum alloy A319 with a pulsed electron beam it is possible to form in the surface layer a structure of the fine dendritic cell (columnar) type without the inclusion of secondary phases of needle (lamellar) shape, which leads to a significant increase in the plasticity of the material with a small change in the strength properties. The depth of the processed zone varies from 15 to 100 µm. The greatest depth of the surface layer is characteristic of samples processed at 25 J/cm2. However, the greatest ductility is characteristic of the samples processed with an electron beam energy of 15 J/cm2.

The data obtained testify to the localization of strain in all types of specimens during testing. This process begins with the formation of a chaotic or staggered distribution of localized areas of plastic deformation. Further development of the plastic flow leads to the formation of strain bands of individual concentrators located at an angle of 45 degrees to the deformation axis. Bands of two orientations are present in the specimens and intersect each other. At the intersection of such bands, areas of maximum localization of plastic deformation are formed. The tendency to band formation increases after surface processing with electron beam, and the deformation localization is maximal in specimens processed with electron beam of energy 15 J/cm2, while the deformation bands are most pronounced in specimens processed at 25 J/cm2. Plastic deformation localization zones can be of stable, metastable and unstable types. Zones of the first type form from small amounts of deformation and develop throughout the test. Zones of the second type may develop over a long period of time, but their effects then diminish or cease. Third type zones form for a short period of time and then disappear. The fracture structure of the base metal of processed and unprocessed specimens is represented by brittle chipping along the boundaries of the silicon inclusions. The destruction of the processed surface layer is characterized by intergranular fracture along the boundaries of columnar dendrite branches.

Thus, the results obtained demonstrate the effectiveness of surface processing of aluminum-silicon alloy products to reduce their brittleness and increase their ductility.

  1. The reference style does not meet the requirements of the journal. DOI is also missing.

A: References to literature have been corrected

Round 2

Reviewer 2 Report

Corrected paper deals with an interesting topic and can be interesting for readers of Materials journal.

Referring to my substantive reservations – the authors made the necessary modifications. They changed the text of the article and removed stylistic and grammatical errors. There are still many confusing sentences with a poor clarity, these making the paper difficult to be read. I am asking for corrections by a native speaker.

Author reformatted and extended body text of a whole paper. Author significantly reformat the entire article. They modified a list of a references by including the most recent relevant publications published in last years.

The paper can be accepted for publication in of Materials journal after languages corrections.

Reviewer 4 Report

The authors have carefully addressed the reviewer's recommendations.